# Prognostic Role of Pre- and Post-Treatment [18F]FDG PET/CT in Squamous Cell Carcinoma of the Oropharynx in Patients Treated with Chemotherapy and Radiotherapy

**DOI:** 10.3390/medsci12030036

**Published:** 2024-07-29

**Authors:** Francesco Dondi, Maria Gazzilli, Domenico Albano, Alessio Rizzo, Giorgio Treglia, Antonio Rosario Pisani, Carmen Palumbo, Dino Rubini, Manuela Racca, Giuseppe Rubini, Francesco Bertagna

**Affiliations:** 1Nuclear Medicine, Università degli Studi di Brescia and ASST Spedali Civili di Brescia, 25123 Brescia, Italy; domenico.albano@unibs.it (D.A.); francesco.bertagna@unibs.it (F.B.); 2Nuclear Medicine, ASL Bari—P.O. Di Venere, 70012 Bari, Italy; marinagazzilli@msn.com; 3Department of Nuclear Medicine, Candiolo Cancer Institute, FPO—IRCCS, 10060 Turin, Italy; alessio.rizzo@ircc.it (A.R.); manuela.racca@ircc.it (M.R.); 4Clinic of Nuclear Medicine, Imaging Institute of Southern Switzerland, Ente Ospedaliero Cantonale, 6501 Bellinzona, Switzerland; giorgio.treglia@eoc.ch; 5Faculty of Biology and Medicine, University of Lausanne, 1011 Lausanne, Switzerland; 6Faculty of Biomedical Sciences, Università della Svizzera Italiana, 6900 Lugano, Switzerland; 7Nuclear Medicine Section, Interdisciplinary Department of Medicine, University of Bari “Aldo Moro”, 70124 Bari, Italy; antoniorosario.pisani@uniba.it (A.R.P.); carmen_med@hotmail.it (C.P.); giuseppe.rubini@uniba.it (G.R.); 8Department of Precision Medicine, University of Campania “L. Vanvitelli”, 80138 Naples, Italy; dinoru95@hotmail.it

**Keywords:** [18]F-FDG, positron emission tomography, PET/CT, oropharynx, head and neck, squamous cell carcinoma, SCC, HPV

## Abstract

Background: The prognostic role of imaging with [18F]fluorodeoxyglucose ([18F]FDG) positron emission tomography/computed tomography (PET/CT) in oropharynx cancer (OPC) has been demonstrated in the past. The aim of this study was to assess the prognostic impact of both baseline and post-treatment PET/CT in patients with OPC and treated with chemo- and/or radiotherapy. Methods: The PET/CT parameters of scans performed before and after therapy were collected and analyzed to find significant prognosticators for progression-free survival (PFS) and overall survival (OS). Human papillomavirus (HPV) infection’s influence on the prognosis was also taken into account. Results: A total of 66 patients were included in the study. The staging volumetric parameters of PET/CT were significant prognosticators for OS, while the same parameters were affordable predictors for PFS at the restaging evaluation. No significant correlations between HPV infection and PET/CT parameters were reported. Conclusion: The prognostic role of volumetric [18F]FDG PET/CT parameters in patients with OPC was reported.

## 1. Introduction

Head and neck squamous cell carcinoma (SCC) is the sixth most common cancer worldwide, with an incidence that is continuously rising, and in this scenario, SCC of the pharynx is the most frequent malignancy of the head and neck region [1,2,3]. The oropharynx is one of the main anatomical parts that compose the pharynx, and the incidence of oropharyngeal cancer (OPC), which is mainly constituted of SCC, is showing an overall increase that is largely attributable to the rise in infections by the human papillomavirus (HPV). In this setting, different risk factors that can contribute to the development of the disease other than HPV infection can be recognized, such as tobacco and alcohol consumption [3,4,5]. The clinical presentation of OPC can be really heterogeneous, with symptoms that are often absent in early stages, while more advanced disease can present with pain, dysphagia, otitis, weight loss, fixation, trismus, paresthesia, and anesthesia. Additionally, in advanced stages, the tumor can metastasize to submandibular, cervical, and jugular lymphatic nodes, and distant metastases most commonly target the lung [5]. Despite that, compared to other forms of head and neck cancers, neoplasms arising in the oral cavity are often diagnosed at early stages due to the presence of mass lesions and symptoms that interfere with eating and speaking [3].

In therapeutic terms, the treatment approach to every patient is guided by anatomical localization, stage, disease characteristics, functional considerations, and patient wishes. Surgical excision is the preferred modality for most well-defined and accessible OPC; however, its use to manage inaccessible or advanced tumors is limited, and in this case, non-surgical treatments, such as chemotherapy (ChT), radiotherapy (RT), or combinations of both these approaches, can be used. For these specific therapies, it is therefore particularly important to predict treatment outcome in order to tailor patient management and subsequent follow-up [5,6,7].

The diagnosis of OPC is based particularly on conventional imaging (CI), with computed tomography (CT), magnetic resonance imaging (MRI), and ultrasound (US) playing pivotal roles in the clear definition of neoplasms’ characteristics and their parameters, such as the node size, internal architecture, and contrast enhancement pattern [6,8]. [18F]fluorodeoxyglucose ([18F]FDG) positron emission tomography/CT (PET/CT) is an imaging modality that in recent years has proved its role in the assessment of several different neoplastic and non-neoplastic conditions [9,10]. The isotope 18F, part of the 18F-FDG molecule, decays by emitting positrons, which annihilate when they combine with electrons. As a result, two gamma ray photons of 511 keV each and at 180 degrees are produced from this electron–positron annihilation. The PET detectors have the ability to detect these photons and, after several steps, to produce an image of the distribution of the radiotracer inside the body of the patient [11]. In this setting, head and neck SCC has been clearly studied with PET/CT imaging that revealed its role for the staging and follow-up of disease [12,13,14,15,16,17]. Compared to other imaging modalities, 18F-FDG PET/CT has the ability to clearly assess the metabolic activity of the primary lesion, aiming therefore to evaluate the amount of vital tissue that is present at the moment of the staging of the disease. In addition, being a whole-body imaging technique, it allows the assessment of the possible presence of local and distant metastatic disease in the staging setting, with a clear impact on the best therapeutic strategy to offer to the patient and, subsequently, on the prognosis. Furthermore, it has been reported that [18F]FDG PET could be an accurate method to decide whether to perform neck dissection in the presence of residual disease after primary ChT and RT in the case of OPC [15]. Several studies have investigated the prognostic role of PET/CT semiquantitative parameters and their role in the prediction of treatment outcomes in OPC. However, in those reports, the diagnostic power was not sufficient, and a certain degree of overlap was reported between patients with good prognoses and those with poor ones [18,19]. Furthermore, the prognostic role of PET/CT imaging is constantly under investigation, since new methodologies, such as texture analysis and radiomics, are emerging and need, therefore, to be clearly evaluated [6,20]. In this setting, hybrid PET/CT imaging has the unique ability to quantify the rate of tracer uptake of a neoplastic lesion, aiming, therefore, at the characterization of its metabolic activity. Different semiquantitative parameters can be extracted from this imaging modality, with the maximum standardized uptake value (SUVmax) reflecting the degree of uptake normalized with the total amount of injected tracer and being one of the most studied parameters since it has the ability to reflect the glycolytic activity of a tumoral lesion. In addition, volumetric parameters, such as metabolic tumor volume (MTV) and total lesion glycolysis (TLG), can be derived from [18F]-FDG imaging: these two semiquantitative parameters are numerical indices that can be extracted from PET/CT images and reflect the total volume of metabolic active disease and the total amount of glycolytic activity of the disease, respectively, and they can be easily calculated with automated contouring methods. It has been recently pointed out that high pre-treatment MTV and/or TLG can predict poor treatment outcomes for HPV-negative OPC patients, while evidence is conflicting for HPV-negative subjects in whom a high baseline metabolic burden is not associated with treatment outcomes [21]. These two semiquantitative parameters are numerical indices that can be extracted from PET/CT images and reflect the total volume of disease and the total amount of glycolytic activity of the disease, respectively.

The aim of this study was therefore to assess the prognostic role of baseline and restaging [18F]FDG PET/CT parameters in OPC SCC in patients treated with ChT and RT.

## 2. Materials and Methods

### 2.1. Patient Selection

To retrieve patients suitable for inclusion in the present study, we retrospectively screened the databases of two institutions, searching for subjects admitted to our centers to perform [18F]FDG PET/CT for the initial staging of OPC. The screening was performed to cover January 2014 to December 2023. The inclusion criteria were the presence of a histologically proven diagnosis of OPC, treatment of the disease based on ChT and RT, and the presence of a baseline [18F]FDG PET/CT scan performed before any therapy. The exclusion criterion was the presence of other neoplastic conditions at the time of the scan. Overall, the final cohort of the study was composed of 66 patients. The observation period was 8 years.

Informed consent was obtained from all individual participants included in the study. Information about gender, age, HPV infection status, the therapy performed, and the American Joint Commission on Cancer (AJCC) VIIIth Edition stage were collected. The presence of HPV infection was assessed by searching for HPV DNA on molecular assay tests.

### 2.2. [18F]FDG PET/CT Acquisition and Interpretation

To perform the [18F]FDG PET/CT scans, the patients fasted for at least 6 h before the exam and had a glucose blood level below 150 mg/dL. Furthermore, 3.5–4.5 MBq/kg of radiotracer was intravenously injected into the patients, and before the image acquisition, they were instructed to void. No intestinal preparation with purge, enteric contrast, or contrast agents was used. At the first center, 60 min after the [18F]FDG injection, images were acquired from the vertex to the midthigh on a Discovery ST or Discovery 690 PET/CT tomograph (General Electric Company, GE, Milwaukee, WI, USA) with standard parameters (CT: 80 mA, 120 kV; PET: 2.5–4 min per bed position, PET step of 15 cm). Reconstruction was performed with a 256 × 256 matrix and a 60 cm field of view. On the Discovery 690 tomograph, time-of-flight (TOF) and point-spread function (PSF) algorithms were used for the reconstruction of the images, with a filter cut-off of 5 mm, 18 subsets, and three iterations. For the Discovery ST tomograph, an ordered subset expectation maximization (OSEM) algorithm with a filter cut-off of 5 mm, 21 subsets, and two iterations was applied. At the second center, 60 min after the [18F]FDG injection, images were acquired from the vertex to the midthigh on a Discovery 710 PET/CT tomograph (General Electric Company, GE, Milwaukee, WI, USA) with standard parameters (CT: 140 mA, 120 kV; PET: 2.3–4 min per bed position, PET step of 15 cm). Reconstruction was performed with a 256 × 256 matrix and a 70 cm field of view. On the Discovery 710 tomograph, TOF and PSF algorithms were used for the reconstruction of the images, with a filter cut-off of 5 mm, 18 subsets, and three iterations.

The PET/CT images were visually and semiquantitatively analyzed by two experienced nuclear medicine physicians by consensus in both institutions, and, for this purpose, every focal tracer uptake deviating from the physiological distribution and from the background was regarded as suggestive of disease localization. Semiquantitative analysis of the images was performed by measuring the SUVmax, MTV, and TLG of the hypermetabolic lesions. In this setting, to calculate the MTV and TLG, an SUV-based automated contouring program (Advantage Workstation 4.6, GE HealthCare, Seoul, Republic of Korea) was used, using an isocontouring threshold method based on 41% of the SUVmax, as recommended by the European Association of Nuclear Medicine [22]. Furthermore, the TLG was calculated as the sum of the product of the MTV of each lesion and its SUVmean. The SUVmax of the liver was calculated using a spheric volume of interest (VOI) with a 1 cm diameter placed at the VIII hepatic segment from the transaxial PET images. A similar VOI was used to obtain the SUVmax of the blood pool at the aortic arch from the transaxial PET images, paying attention to not involve the vessel’s walls. These two values were used to calculate the ratio between the highest SUVmax of the neoplastic lesions (SL and SBP, respectively). The SUVmax was assessed at the point with the highest uptake, considering both primary and metastatic lesions, and the volumetric parameters, such as MTV and TLG, were extracted by definition considering all the lesions.

### 2.3. Statistical Analysis

MedCalc Software version 18.1 for Windows (Ostend, Belgium) was used to perform all statistical analyses. Descriptive analysis of the categorical variables was carried out comprising the calculation of the simple and relative frequencies. Moreover, the numeric variables were described as mean, standard deviation (SD), minimum, and maximum values (range). A *t*-test was used to evaluate the presence of differences in terms of the semiquantitative PET/CT parameters and the clinico-pathological features of the patients. To estimate the survival rate and the risk of disease progression, overall survival (OS) and progression-free survival (PFS) were calculated. In particular, OS was defined as the time in months from the date of the baseline [18F]FDG PET/CT scan to the date of death from any cause or to the date of the last documented follow-up. PFS was calculated as the time in months between the baseline [18F]FDG PET/CT scan and the date of the first documented relapse or disease progression, based on radiological imaging (CT, MRI and PET/CT) and/or biopsy results.

A Cox regression model was applied to identify independent prognosticators between the clinicopathological, PET/CT, and MRI features. This analysis was performed for both the staging and restaging PET/CT scans. To do that, PET/CT semiquantitative parameters were dichotomized based on their median value, as was made also for age. Stage was dichotomized between stages I–II and III–IV. Estimates of the predictive effect for PFS and OS were expressed as hazard ratios (HRs) in univariate and multivariate Cox regression analyses with a 95% confidence interval (CI). For all the aforementioned statistics, a *p*-value < 0.05 was considered significant.

A Kaplan–Meier analysis was used to draw survival curves, and a log-rank test was then applied to compare these curves, again considering a *p*-value < 0.05 as statistically significant. These analyses were performed again to dichotomize the PET/CT semiquantitative parameters on their median value.

## 3. Results

The total cohort of the study was composed of 66 subjects; 45 of them were men (68.2%). The mean age was 51 years (SD 11, range 38–86) (Table 1). Eighteen (27.3%) patients were diabetics. Data regarding the presence of HPV infection were available only for 46 patients: 19 (41.3%) were positive for the infection, while 27 (58.7%) were negative. According to the VIIIth edition of the AJCC staging system, two patients (3.0%) had stage I disease, 24 (36.4%) had stage II disease, 36 (54.5%) had stage III disease, while four (6.1%) subjects had stage IV disease. In this setting, the presence of nodal metastases at diagnosis was reported in 61 patients (92.4%) while the remaining five subjects (7.6%) did not demonstrate nodal localization of disease (Figure 1). Furthermore, distant metastases were present at diagnosis in four patients (6.1%). A positive restaging PET/CT scan was available for 34 subjects only. The mean PFS of our cohort was 31.7 months (SD 28.1, range 3.0–89.1), and relapse or progression of disease were experienced by 31 patients (47.0%). Additionally, the mean OS was 37.1 months (SD 25.9, range 3.0–89.1), and death occurred in 11 patients (16.7%). 

Overall, for most of the clinico-pathological features of the patients, no significant differences in terms of the semiquantitative PET/CT parameters were reported. An exception was the difference in terms of the SUVmax, SL, MTV, and TLG between the patients with stages I–II and the subjects with stages III–IV. In addition, statistically different SUVmax values were reported between the patients with or without nodal and distant metastasis, and also the SL was different for the patients with or without nodal metastasis. In this setting, it is, however, worth underlining that the patients without nodal metastasis and the patients with distant metastasis were small samples (Table 2).

The Kaplan–Meier analyses revealed the staging PET/CT semiquantitative parameters as predictive for both PFS and OS, with the exception of SL for both of them and TLG for PFS. When considering the restaging scans, again, most of the PET/CT parameters confirmed their prognostic value, with the exception of SL and TLG for OS (Table 3 and Figure 2).

After performing univariate analyses of the staging parameters stage, SUVmax, SBP, and MTV were demonstrated as predictors of PFS. In the subsequent multivariate analyses, from which SBP was excluded, since it is mathematically derived from SUVmax, no independent predictors were confirmed. For OS, the univariate analyses revealed SUVmax, SBP, MTV, and TLG as affordable prognosticators, while the subsequent multivariate analyses, again excluding SBP and TLG for the reason mentioned before, confirmed only MTV as a significant independent predictor. Focusing on the restaging scans, all PET/CT semiquantitative parameters were reported as significant prognosticators for PFS in the univariate analyses, while the subsequent multivariate analyses confirmed only MTV as an affordable independent predictor. Additionally, the univariate investigation for OS revealed SUVmax, SBP, and MTV as prognosticators, while in the subsequent multivariate analyses, none of them were confirmed as affordable independent predictors (Table 4 and Table 5).

## 4. Discussion

In order to tailor therapy, subsequent management, and follow-up, it is important to find suitable prognostic predictors in patients with OPC. Imaging assessment of the disease could provide fundamental information, and the potential role of [18F]FDG PET/CT to give prognostic information has emerged in the past [18]. Furthermore, staging and restaging assessments of the disease after therapy with PET imaging have different and particular characteristics and impacts that need to be clearly evaluated and taken into account.

Our results confirmed the potential prognostic role of [18F]FDG PET/CT in SCC OPC. In particular, in the staging scans, no significant predictors were reported for PFS, while volumetric parameters (MTV and TLG) were significant prognosticators for OS. Since these two parameters reflect the total amount of disease in volumetric and metabolic indices, respectively, it is not surprising that patients with a higher quantity of vital disease, which can also reflect the possible presence of metastatic disease, are characterized by the worst prognosis. The value of baseline PET/CT for the prediction of prognosis and the correct setup of patients’ therapeutic management has been also studied in the past. In this setting, its prognostic role before surgery and postoperative RT in OPC has been reported in the past by Choi et al. [19], revealing associations of volumetric parameters with survival, recurrence, invasion depth, and extranodal extension. Similarly, it has also been revealed that SUVmax could be a useful tool for the preoperative evaluation of neck nodal metastasis in patients with SCC of the pharynx and larynx, however with insufficient sensitivity [8]. Interestingly, radiomics and texture analysis performed at staging have more recently demonstrated the ability to extract some features that are potentially predictors of disease progression in patients with pharynx SCC, therefore becoming potential tools to determine the need for additional treatments or follow-up strategies [6,20].

Different to the findings for the staging scans, our findings confirmed restaging PET/CT scans’ volumetric parameters as prognosticators for PFS, while no affordable predictors were revealed for OS in our cohort. As previously underlined in the analyses for the staging scans, these parameters reflected the total amount of disease and the fact that they were prognosticators for PFS. Interestingly, no impact was found on OS, and a possible explanation of this fact could be researched in more specific and tailored therapy resulting from this imaging procedure. The prognostic role of post-treatment [18F]FDG PET/CT was investigated by Urban et al. [23], who demonstrated its clinical usefulness in the assessment of treatment response in oropharynx SCC, revealing also the fact that it could be considered a clinically relevant prognostic factor for complete metabolic response, therefore predicting improved OS in patients with both p16+ or p16- status. Similarly, it has also been reported that in SCC OPC, one of the significant predictors of outcome is the post-treatment SUVmax of the primary site [18]. Our data therefore strengthen the role of PET imaging in the restaging of OPC after therapy, even though we reported a role only for the prediction of PFS.

Different parameters were reported as prognostic factors in the univariate analyses, while they were not confirmed as independent prognosticators in the multivariate analyses. This was the case for the SUVmax and most of the SUV-related ratios in both the staging and restaging PET/CT scans. In general, they reflect the higher metabolic activity of the tumor and can somehow be considered as confounding factors in these analyses, since they are comprised in the calculation of both MTV and TLG, as previously underlined. In addition, we have mentioned that volumetric parameters reflect the total amount of disease in volumetric and metabolic values; therefore, it is not surprising that these are parameters that more completely reflect the disease and that are therefore able to influence the prognoses of these patients. Similar considerations can be made for the PFS analysis in the staging setting, where the stage of the disease was also underlined as a prognosticator only in the univariate analysis.

Our analyses revealed a significant prognostic role for [18F]FDG-based volumetric parameters, such as MTV and TLG, in the setting of SCC OPC treated with chemo- and radiotherapy. As mentioned before, these parameters reflect the total amount of volumetric active disease and its total glycolytic activity; therefore, our results are not surprising. From a clinical point of view, we can hypothesize that these two parameters can reflect some particular aspects of the disease. Since MTV is calculated by the sum of all tumoral lesions with a threshold of 41% of the SUVmax, this value is strictly correlated to the volume of disease, and, therefore, more extended tumors with the presence of nodal or distant metastases are characterized by the highest values of MTV. Similarly, TLG is calculated as the product of the MTV of each neoplastic lesion with the mean value of its SUV, and, therefore, this parameter, as the name suggests, reflects the total glycolytic activity of the disease, considering both the volume and intensity of uptake. It is not surprising that neoplasms with more aggressive behaviors, therefore with a higher glycolytic activity and consequently a higher intensity of tracer uptake, and/or characterized by the presence of nodal and distant metastases, have a higher TLG and worst prognosis. In addition, both MTV and TLG are numerical indices that can be easily extracted from [18F]FDG PET/CT imaging with SUV-based automated contouring programs as mentioned, and, therefore, they could be of particular importance in the correct assessment of OPC SCC patients, giving some additional information on the status of the disease, aiming for more clear and patient-centered therapeutic or follow-up regimens. Lastly, it is worth underlining that this metabolic information can be extracted only from [18F]FDG imaging and that PET/CT is a whole-body technique with the ability, therefore, to assess the presence of disease in the whole body of the patients, with a clear advantage when setting up a specific therapeutic approach.

Several studies have reported the value of [18F]FDG PET/CT for the screening of the presence of metastases in head and neck cancer [24]. Additionally, it has been demonstrated that the presence of metastases is a major determinant for both the management and prognosis of SCC of this anatomical region [25,26]. Extra-nodal tumor spread and high lymph node metastatic burdens are, moreover, known to contribute to poor treatment outcomes in surgically treated HPV-positive OPC or oral cancer patients [21]. In this setting, it has been proposed that, for PET/CT, patients diagnosed with distant metastases at initial screening have significantly worse expected survival compared to the group diagnosed during follow-up [27]. In our cohort, volumetric parameters, which reflect the total volumetric burden of disease and the total metabolic burden of the neoplasm, were correlated with OS and PFS. This finding confirms, therefore, the aforementioned insights.

A recent and wide comprehensive review revealed that a high pre-treatment metabolic burden before ChT/RT can predict poor treatment outcomes for HPV-negative OPC patients, while evidence is conflicting in the case of HPV-positive OPC patients. Additionally, emerging data to support the use of nodal metabolic and/or mid-treatment response parameters for HPV-positive patients are emerging [21]. Interestingly, we did not report any significant difference in PET/CT semiquantitative parameters between the patients with a documented HPV infection and subjects without the infection, and, in addition, we did not find influences of HPV infection on prognosis.

Some important limitations affect our work and need to be considered in order to interpret our results. First of all is the retrospective design of the study. In addition, the total cohort was composed of a relatively limited sample. Moreover, only 11 patients (16.7%) died during the follow-up, therefore weakening our findings for this specific analysis. In addition, the value of the assessment for correlation with HPV infection and the prognostic role of post-treatment [18F]FDG PET/CT was weakened by the low samples considered. Lastly, [18F]FDG PET/CT is not routinely performed before and after treatment in all patients with OPC SCC; therefore, hidden selection bias deriving from this fact could be present.

## 5. Conclusions

We confirmed the prognostic role of volumetric [18F]FDG PET/CT parameters in OPC SCC treated with ChT and/or RT, in particular for baseline scans with OS and post-treatment scans with PFS.

## Figures and Tables

**Figure 1 medsci-12-00036-f001:**
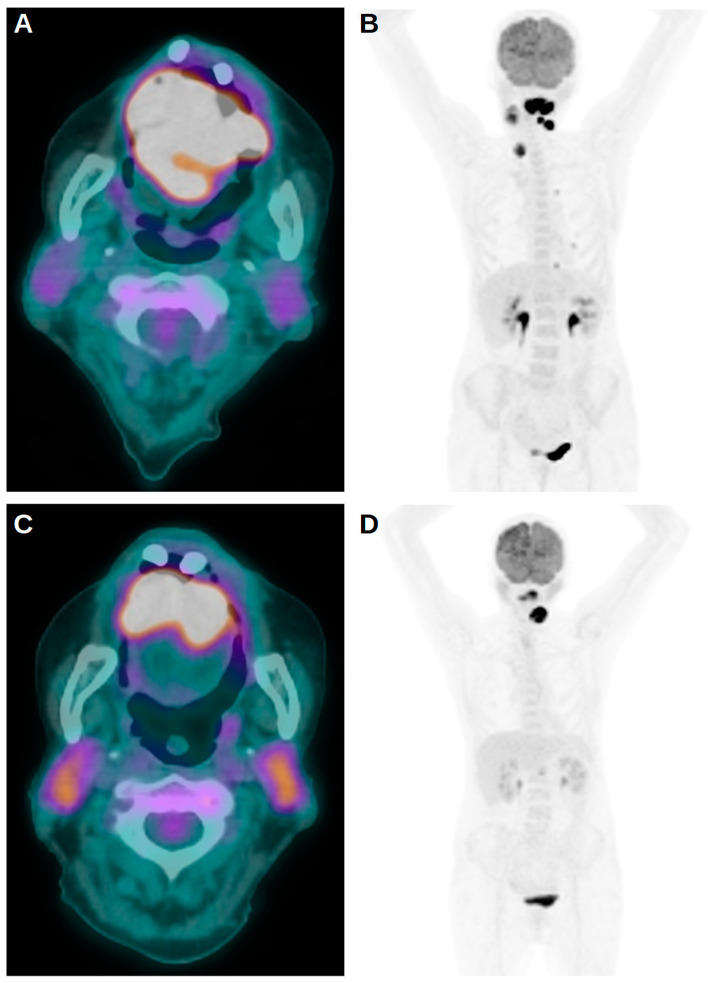
Fused PET/CT (**A**) and maximum index projection (MIP) (**B**) images of an [18F]FDG scan performed for staging purposes in a patient with an SCC of the oropharynx. The exam demonstrated the presence of a primary tumor but also of nodal and lung metastases. After undergoing ChT and RT, the subject again underwent a PET/CT scan 5 months later (**C**,**D**) that revealed the persistence of local and nodal uptake of the tracer. Four months afterwards, the relapse of disease on the lung and the progression on nodes were demonstrated, and the patient died 2 months later.

**Figure 2 medsci-12-00036-f002:**
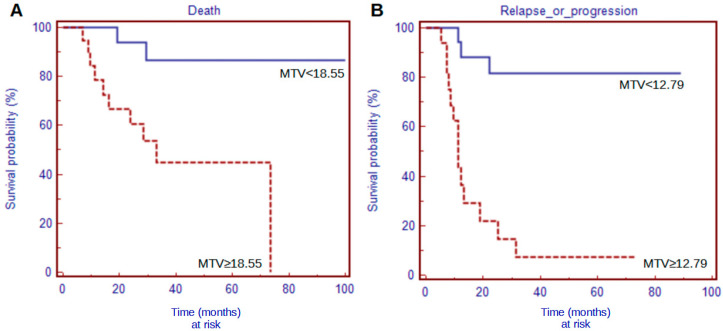
Representative survival curves for OS with baseline MTV(**A**) and for PFS with restaging MTV (**B**).

**Table 1 medsci-12-00036-t001:** Characteristics of the 66 patients included in the study.

Characteristic	Number (%)
Sex	
Male	45 (68.2)
Female	21 (31.8)
Age (mean ± SD, range)	51 ± 11, 38–86
AJCC stage	
I	2 (3.0)
II	24 (36.4)
III	36 (54.5)
IV	4 (6.1)
Nodal metastases at diagnosis	
Yes	61 (92.4)
No	5 (7.6)
Distant metastases at diagnosis	
Yes	4 (6.1)
No	62 (93.9)
HPV	
Positive	19 (28.8)
Negative	27 (40.9)
Not available	20 (30.3)
PET/CT parameters	
Staging	
SUVmax (mean ± SD, range)	16.3 ± 7.7, 4.8–41.6
SL (mean ± SD, range)	6.0 ± 3.0, 1.5–17.1
SBP (mean ± SD, range)	7.6 ± 3.9, 1.6–22.9
MTV (mean ± SD, range)	21.9 ± 14.54, 3.0–75.9
TLG (mean ± SD, range)	292.6 ± 289.4, 3.0–1314.0
Restaging *	
SUVmax (mean ± SD, range)	8.4 ± 8.3, 2.1–48.7
SL (mean ± SD, range)	3.6 ± 4.8, 0.9–28.6
SBP (mean ± SD, range)	4.9 ± 7.6, 1.2–45.6
MTV (mean ± SD, range)	18.1 ± 19.6, 2.2–100.6
TLG (mean ± SD, range)	288.8 ± 609.5, 3.9–3371.6
Relapse or progression	
Yes	31 (47.0)
No	35 (53.0)
Death	
Yes	11 (16.7)
No	55 (83.3)
PFS months (mean ± SD, range)	31.7 ± 28.1, 3.0–89.1
OS months (mean ± SD, range)	37.1 ± 25.9, 3.0–89.1

* Data available only for 34 patients. SD: standard deviation; AJCC: American Joint Commission on Cancer; PET/CT: positron emission tomography/computed tomography; SUVmax: standardized uptake value body weight max; SL: SUVmax/liver uptake; SBP: SUVmax/blood pool uptake; MTV: metabolic tumor volume; TLG: total lesion glycolysis; PFS: progression-free survival; OS: overall survival; MRI: magnetic resonance imaging.

**Table 2 medsci-12-00036-t002:** Difference in mean PET/CT semiquantitative parameters related to different clinico-pathological conditions.

Parameter	SUVmax	*p*-Value	SBP	*p*-Value	SL	*p*-Value	MTV	*p*-Value	TLG	*p*-Value
SUVmax		0.016		65		0.033		0.037		0.007
SBP	13.5		6.52		5.09		17.32		175.8	
Stage	18.2		8.33		6.68		24.89		368.5	
1–2		0.479		0.964		0.864		0.335		0.74
3–4	16		7.6		5.9		23.7		304.9	
Age	16.6		7.6		6.1		20.2		281	
<62		0.575		0.686		0.971		0.452		0.718
≥62	17.1		7.9		6.1		19.9		273.6	
Sex	15.9		7.5		6		22.8		301.5	
Male		0.018		0.084		0.01		0.177		0.167
Female	8.6		4.7		3.1		13.4		120	
Nodal metastases	16.9		7.9		6.3		22.6		306.7	
No		0.039		0.069		0.442		0.793		0.252
Yes	15.8		7.4		5.9		21.8		282.2	
Distant metastases	23.9		11		7.1		23.8		454.2	
No		0.462		0.938		0.804		0.326		0.797
Yes	15.7		7.8		6.1		22.3		310.9	
HPV infection	17.5		7.9		6.4		21.9		308.2	
Negative										
Positive										

SUVmax: standardized uptake value body weight max; SL: SUVmax/liver uptake; SBP: SUVmax/blood pool uptake; MTV: metabolic tumor volume; TLG: total lesion glycolysis; PFS: progression-free survival; OS: overall survival; PET/CT: positron emission tomography/computed tomography; HPV: human papillomavirus.

**Table 3 medsci-12-00036-t003:** Kaplan–Meier results (*p*-value) of PET/CT semiquantitative parameters for PFS and OS.

Parameter	3 Years PFS	5 Years PFS	*p*-Value	3 Years OS	5 Years OS	*p*-Value
Staging			0.029			0.012
SUVmax						
<16.1	79%	68%		92%	89%	
≥16.1	54%	49%	0.013	69%	68%	0.018
SBP						
<7.2	79%	79%		95%	95%	
≥7.2	52%	47%	0.051	74%	66%	0.074
SL						
<5.7	81%	64%		88%	76%	
≥5.7	72%	59%	0.016	76%	59%	0.01
MTV						
<18.6	84%	84%		87%	87%	
≥18.6	53%	41%	0.156	59%	43%	0.006
TLG						
<203.4	83%	71%		86%	85%	
≥203.4	66%	62%		60%	54%	0.037
Restaging			0.015			
SUVmax						
<5.4	77%	75%		76%	76%	0.037
≥5.4	59%	50%	0.015	55%	51%	
SBP						
<2.9	78%	78%		77%	73%	0.077
≥2.9	43%	32%	0.006	58%	52%	
SL						
<2.2	82%	74%		79%	79%	0.014
≥2.2	48%	31%	<0.001	61%	60%	
MTV						
<12.8	81%	81%		84%	80%	0.066
≥12.8	16%	9%	0.008	53%	53%	
TLG						
<95.1	81%	81%		82%	80%	
≥95.1	18%	10%		49%	41%	

SUVmax: standardized uptake value body weight max; SL: SUVmax/liver uptake; SBP: SUVmax/blood pool uptake; MTV: metabolic tumor volume; TLG: total lesion glycolysis; PFS: progression-free survival; OS: overall survival.

**Table 4 medsci-12-00036-t004:** Univariate and multivariate analyses of clinico-pathological and [18F]FDG PET/CT semiquantitative parameters for staging parameters.

	Univariate Analysis		Multivariate Analysis	
	PFS			
	*p*-value	HR (95% CI)	*p*-value	HR (95% CI)
Stage	0.024	2.29 (1.07–4.90)	0.053	2.16 (0.99–4.70)
Age	0.228	1.52 (0.76–3.03)		
Sex	0.112	1.85 (0.83–4.09)		
HPV	0.884	1.05 (0.53–2.06)		
SUVmax	0.03	2.12 (1.06–4.23)	0.401	1.38 (0.65–2.94)
SBP	0.013	2.37 (1.17–4.79)		
SL	0.052	1.96 (0.98–3.91)		
MTV	0.017	2.29 (1.14–4.59)	0.06	2.05 (0.97–4.32)
TLG	0.159	1.63 (0.82–3.24)		
	OS			
Stage	0.688	1.22 (0.44–3.36)		
Age	0.68	0.81 (0.30–2.16)		
Sex	0.912	2.18 (0.62–7.70)		
HPV	0.353	1.59 (0.59–4.26)		2.23 (0.67–7.33)
SVmax	0.017	3.84 (1.24–11.89)	0.188	
SBP	0.017	3.57 (1.15–11.04)		
SL	0.072	2.53 (0.88–7.26)		5.25 (1.32–20.78)
MTV	<0.001	7.27 (1.93–27.32)	0.018	
TLG	0.007	4.41 (1.42–13.63)		

SUVmax: standardized uptake value body weight max; SL: SUVmax/liver uptake; SBP: SUVmax/blood pool uptake; MTV: metabolic tumor volume; TLG: total lesion glycolysis; PFS: progression-free survival; OS: overall survival; HR: hazard ratio; CI: confidence interval; HPV: human papillomavirus infection.

**Table 5 medsci-12-00036-t005:** Univariate and multivariate analyses of clinico-pathological and [18F]FDG PET/CT semiquantitative parameters for restaging parameters.

	Univariate Analysis		Multivariate Analysis	
	PFS			
	*p*-value	HR (95% CI)	*p*-value	HR (95% CI)
Stage	0.091	2.36 (0.83–6.71)		
Age	0.51	0.71 (0.27–1.88)		
Sex	0.055	3.03 (0.86–10.59)		
HPV	0.599	1.38 (0.42–4.47)		
SVmax	0.017	3.32 (1.20–9.14)	0.553	1.39 (0.46–4.19)
SBP	0.017	3.32 (1.20–9.14)		
SL	0.001	3.99 (1.39–11.43)		
MTV	<0.001	10.85 (3.07–38.32)	0.001	9.47 (2.47–36.25)
TLG	0.009	3.74 (1.31–10.68)		
	OS			
Stage	0.601	1.46 (0.34–610)		
Age	0.368	0.52 (0.13–2.09)		
Sex	0.063	5.83 (0.64–52.84)		
HPV	0.739	0.70 (0.08–5.99)		
SUVmax	0.039	4.69 (0.95–23.15)	0.407	2.23 (0.33–14.79)
SBP	0.039	4.69 (0.95–23.15)		
SL	0.075	3.81 (0.77–18.79)		
MTV	0.017	6.05 (1.20–30.43)	0.17	3.80 (0.56–25.49)
TLG	0.064	4.01 (0.81–19.74)		

SUVmax: standardized uptake value body weight max; SL: SUVmax/liver uptake; SBP: SUVmax/blood pool uptake; MTV: metabolic tumor volume; TLG: total lesion glycolysis; PFS: progression-free survival; OS: overall survival; HR: hazard ratio; CI: confidence interval; HPV: human papillomavirus infection.

## Data Availability

Data available on request due to privacy/ethical restrictions.

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
