# Peer review of "Prognostic Role of Pre- and Post-Treatment [18F]FDG PET/CT in Squamous Cell Carcinoma of the Oropharynx in Patients Treated with Chemotherapy and Radiotherapy"

_medsci, 2024, doi:10.3390/medsci12030036_

Round 1
Reviewer 1 Report
Comments and Suggestions for Authors
The reviewer considers that the following should be addressed, in order to endorse the publication of the manuscript:
1. Introduction section does not provide enough information on potential advantages of [18F] FDG PET/CT in comparison to other imagiological methods to the diagnosis of head/neck SCC. This should be corrected accordingly.
2.Introduction section does not convey details [18F] FDG PET/CT technique, fom a mechanistic point of view. This section should de extended to include details on the following: how [18F] FDG is used to generate signal? and how is must interpreted in clinical context.
3.Line 79. It should be indicated what MTV and TLG stand for, along with details on its relevance.
4. Materials and methods sections. It is not entirely clear if only primary tumor or proximal metasteses were assessed in the acquired datasets with [18F] FDG PET/CT. This should be corrected accordingly.
5. Discussion section should be extended to include a broader discussion of the attained results, regarding the parameters deemed to be prognosis predictors, by relating it with tumoral biochemical/metabolic features.
Comments on the Quality of English LanguageMinor proofreading is required.
Author Response
Dear Reviewer,
thank you for the precise and useful evaluation of the manuscript. We carefully considered all your comments and integrated or corrected the manuscript according to your request.
In particular:
- “Introduction section does not provide enough information on potential advantages of [18F] FDG PET/CT in comparison to other imagiological methods to the diagnosis of head/neck SCC. This should be corrected accordingly.”: a brief integration was added to the introduction;
- “Introduction section does not convey details [18F] FDG PET/CT technique, fom a mechanistic point of view. This section should de extended to include details on the following: how [18F] FDG is used to generate signal? and how is must interpreted in clinical context.”: similarly to the previous point, we added a brief part which focuses on the mechanisms behind positron emission imaging;
- “Line 79. It should be indicated what MTV and TLG stand for, along with details on its relevance.”: these information were added to the introduction;
- “Materials and methods sections. It is not entirely clear if only primary tumor or proximal metasteses were assessed in the acquired datasets with [18F] FDG PET/CT. This should be corrected accordingly.”: we better specified the method used to assess PET/CT semiquantitative parameters;
- “Discussion section should be extended to include a broader discussion of the attained results, regarding the parameters deemed to be prognosis predictors, by relating it with tumoral biochemical/metabolic features”: a brief discussion on this topic has been added;
- the whole text was proofread according to your request.
Best regards,
the authors.
Reviewer 2 Report
Comments and Suggestions for Authors
Thank you for the opportunity to review this manuscript. This study examined the prognostic value of FDG-PET during staging and restaging in oropharyngeal cancer, and the results are very interesting. However, there are some points that should be added and revised before the manuscript can be accepted.
1. Is FDG-PET routinely performed before and after treatment at the two institutions? If not, this should be noted in the limitations, as this suggests hidden selection bias.
2. Please add the observation period.
3. Please provide details of the criteria for determining HPV positivity.
4. Were there any cases of comorbid diabetes? If so, this should be noted in the exclusion criteria.
5. For both staging and restaging, many factors were reflected as having significant prognostic predictive ability in the univariate analysis, whereas only a limited number of items were significant in the multivariate analysis. Please clearly indicate the degree of confounding for each item and provide your commentary.
6. Regarding Table 1:
There is no description of the tumor subsite and treatment method. Please add them.
7. Regarding Table 2:
Please provide parameters that indicate the degree of correlation, not just the p-value.
8. Regarding Table 3:
Please provide 3-year and 5-year OS and PFS, not just the p-value.
9. About Figure 2:
The unit of "time" on the horizontal axis is not written. Also, please include "at risk" below the graph.
Author Response
Dear Reviewer,
thank you for the precise and useful evaluation of the manuscript. We carefully considered all your comments and integrated or corrected the manuscript according to your request.
In particular:
- “Is FDG-PET routinely performed before and after treatment at the two institutions? If not, this should be noted in the limitations, as this suggests hidden selection bias.”: this point was added in the limitation at the end of the discussion;
- “Please add the observation period.”: the observation period was added;
- “Please provide details of the criteria for determining HPV positivity.”: these details were added to the text;
- “Were there any cases of comorbid diabetes? If so, this should be noted in the exclusion criteria.”: the number of patients with diabetes was specified in the text. However this fact was not included in the exclusion criteria since diabetics patients with a good preparation in terms of therapy and with a blood glucose level <150 mg/dL can still perform an 18F-FDG PET/CT scan;
- “For both staging and restaging, many factors were reflected as having significant prognostic predictive ability in the univariate analysis, whereas only a limited number of items were significant in the multivariate analysis. Please clearly indicate the degree of confounding for each item and provide your commentary.”: this part was added and specified in the discussion;
- “Regarding Table 1: There is no description of the tumor subsite and treatment method. Please add them.”: in the inclusion criteria we stated that all the patients were affected by SCC of the oropharynx and that all of them were treated with chemo- and radiotherapy, therefore we think that it is not necessary to add these information to the table. We hope that you can agree with us;
- “Regarding Table 2: Please provide parameters that indicate the degree of correlation, not just the p-value.”: this is a good point, this table was completely modified. First of all this is not a correlation analysis but a T-test, therefore we corrected our previous commentary by removing the word “correlation”. In addition, values of the semiquantitative parameters for HPV positive and negative patients were added to clarify the whole table. Lastly, the whole table was changed and integrated in a new “Table 2” to answer to the requests of another Reviewer;
- “Regarding Table 3: Please provide 3-year and 5-year OS and PFS, not just the p-value.”: these values were added to the table;
- “About Figure 2: The unit of "time" on the horizontal axis is not written. Also, please include "at risk" below the graph.”: the figure was modified according to your requests.
Best regards,
the authors.
Reviewer 3 Report
Comments and Suggestions for Authors
In this study, the authors aimed to assess the prognostic impact of both baseline and post-treatment PET/CT in patients with oropharynx cancer (OPC) and treated with chemo- and/or radio-therapy. They collected and analyzed PET/CT parameters of scans performed before and after therapy to find significant prognosticators for progression free survival (PFS) and overall survival (OS). The authors found that staging volumetric parameters of PET/CT were significant prognosticators for OS, while the same parameters were affordable predictors for PFS at the restaging evaluation.
In overall, this study could guide clinical diagnosis and treatment for OPC patients by indicating prognostic role of [18F] FDG PET/CT. The study design was reasonable. The manuscript was well-presented and expressed clearly. Below are suggestions:
#1. Please provide exclusion criteria in the part of 2.1. Patients selection.
#2. In the lines 199-200, the conclusion of “When considering the restaging scans, again most of the PET/CT parameters confirmed their prognostic value with the exception of SBP and TLG for OS (Table 3 and Figure 2)” was wrong, the p-value of SBP for OS is 0.037, while SL is 0.077. Therefore, the conclusion should be “When considering the restaging scans, again most of the PET/CT parameters confirmed their prognostic value with the exception of SL and TLG for OS (Table 3 and Figure 2)”.
#3. Some statements were not accurate, e.g. “Additionally, univariate investigation for OS revealed SUVmax, SBP and MTV as prognosticators while at subsequent multivariate analysis none of them were confirmed as affordable predictors (Table 4 e Table 5).” in the lines 229-232. Univariate analysis examines each variable independently without considering the relationships between variables, while multivariate analysis examines the relationships between two or more variables simultaneously. If p < 0.05 in multivariate analysis, it means this variable is an independent predictor for patients. In this case, SUVmax, SBP and MTV were not independent predictor, but they were still affordable predictors for OPC patients because their p-value <0.05 in univariate analysis.
#4. To improve this study, the authors can try to analysis the correlation between PET/CT semiquantitative parameters (SUVmax, SBP, SL, MTV and TLG) and clinicopathological characteristics (Stage, Age, Sex, Nodal metastases and Distant metastases).
Author Response
Dear Reviewer,
thank you for the precise and useful evaluation of the manuscript. We carefully considered all your comments and integrated or corrected the manuscript according to your request.
In particular:
- “Please provide exclusion criteria in the part of 2.1. Patients selection.”: exclusion criteria were added to the text;
- “In the lines 199-200, the conclusion of “When considering the restaging scans, again most of the PET/CT parameters confirmed their prognostic value with the exception of SBP and TLG for OS (Table 3 and Figure 2)” was wrong, the p-value of SBP for OS is 0.037, while SL is 0.077. Therefore, the conclusion should be “When considering the restaging scans, again most of the PET/CT parameters confirmed their prognostic value with the exception of SL and TLG for OS (Table 3 and Figure 2)”: the sentence was changed with this suggestion;
- “#3. Some statements were not accurate, [...] patients because their p-value <0.05 in univariate analysis.”: the text of the results session was changed according to these suggestions;
- “To improve this study, the authors can try to analysis the correlation between PET/CT semiquantitative parameters (SUVmax, SBP, SL, MTV and TLG) and clinicopathological characteristics (Stage, Age, Sex, Nodal metastases and Distant metastases).”: this analysis were performed and added to the results section.
Best regards,
the authors.
Round 2
Reviewer 1 Report
Comments and Suggestions for Authors
Upon carefull consideration, the reviewer considers rather insufficient the author´s response to reviewer´s remarks. Therefore, the reviewer considers that the following should be addressed:
The manuscript fails to provide details about this particular imagiological technique allows the calculation of metabolic tumor volume and total lesion glycolysis, and how those two parameters relate to biochemical features of squamous cells carcinoma. This is of utmost importance to establish the advantages of the technique in comparison with other more widespread imagiological diagnostic methods. In this perpective, the introduction and discussion sections should be extended to include a broader discussion in this regard.
Although author´s have included in the introduction section details on mechanistic undrelying the technique, it does not provide details on the generated images shoul be interpreted in clinical context. This should be corrected accordingly.
Author Response
Dear Reviewer,
thank you again for the evaluation of the manuscript. We carefully considered your comment and integrated or corrected the manuscript according to your request. In particular, we extended the introduction and the discussion section of the manuscript to include more details on how MTV and TLG are extracted from 18F-FDG imaging, on their possible correlation with clinical aspects of SCC and how the could possibly be integrated in the assessmen of the disease.